# A multistudy analysis reveals that evoked pain intensity representation is distributed across brain systems

**Bogdan Petre**[1], **Philip Kragel**[2], **Lauren Y. Atlas**[3,4,5], **Stephan Geuter**[6], **Marieke Jepma**[7], **Leonie Koban**[8], **Anjali Krishnan**[9], **Marina Lopez-Sola**[10], **Elizabeth A. Reynolds Losin**[11], **Mathieu Roy**[12], **Choong-Wan Woo**[13], **Tor D. Wager**[1] *

1 Dartmouth College, Hanover, New Hampshire, United States of America, 2 University of Colorado Boulder, Colorado, United States of America, 3 National Center for Complementary and Integrative Health, National Institutes of Health, Bethesda, Maryland, United States of America, 4 National Institute of Mental Health, National Institutes of Health, Bethesda, Maryland, United States of America, 5 National Institute on Drug Abuse, National Institutes of Health, Baltimore, Maryland, United States of America, 6 Johns Hopkins University, Baltimore, Maryland, United States of America, 7 University of Amsterdam, Amsterdam, the Netherlands, 8 INSEAD Fontainebleau & ICM Paris, Paris, 9 Brooklyn College of the City University of New York, Brooklyn, New York, United States of America, 10 Department of Medicine, School of Medicine and Health Sciences, University of Barcelona, Barcelona, Spain, 11 Department of Psychology, University of Miami, Coral Gables, Florida, United States of America, 12 McGill University, Montreal, Quebec, Canada, 13 Center for Neuroscience Imaging Research, Institute for Basic Science, Suwon, Gyeonggi-do, Republic of Korea

* Tor.D.Wager@Dartmouth.edu

**Data Availability Statement:** Data are available as 'single trial' statistical contrast maps, hosted at figshare.com and best accessed via the CANlab Single Trials Repository on github (https://github.

## Abstract

Information is coded in the brain at multiple anatomical scales: locally, distributed across regions and networks, and globally. For pain, the scale of representation has not been formally tested, and quantitative comparisons of pain representations across regions and networks are lacking. In this multistudy analysis of 376 participants across 11 studies, we compared multivariate predictive models to investigate the spatial scale and location of evoked heat pain intensity representation. We compared models based on (a) a single most pain-predictive region or resting-state network; (b) pain-associated cortical–subcortical systems developed from prior literature ("multisystem models"); and (c) a model spanning the full brain. We estimated model accuracy using leave-one-study-out cross-validation (CV; 7 studies) and subsequently validated in 4 independent holdout studies. All spatial scales conveyed information about pain intensity, but distributed, multisystem models predicted pain 20% more accurately than any individual region or network and were more generalizable to multimodal pain (thermal, visceral, and mechanical) and specific to pain. Full brain models showed no predictive advantage over multisystem models. These findings show that multiple cortical and subcortical systems are needed to decode pain intensity, especially heat pain, and that representation of pain experience may not be circumscribed by any elementary region or canonical network. Finally, the learner generalization methods we employ provide a blueprint for evaluating the spatial scale of information in other domains.

com/canlab/canlab_single_trials). One additional dataset is used for validating model specificity to pain vs. non-pain stimuli. That additional dataset can be found here: https://neurovault.org/api/collections/3324/. Finally, all numerical data underlying figures in this manuscript are available through github at https://github.com/canlab/petre_scope_of_pain_representation Appropriate statements regarding dataset access can be found in the Methods: Data section of our main manuscript document, and in each figure legend.

**Funding:** Data storage was supported by the University of Colorado Boulder "PetaLibrary". Research was funded by NIMH R01 MH076136 (TDW), NIDA R01 DA046064 (TDW) and NIDA R01 DA035484 (TDW). Research was additionally supported by funding from NCCIH ZIA AT000030 (LYA), NIDA K01 DA045735 (EARL), and the Serra Hunter fellow lecturer program (ML). The content is solely the responsibility of the authors and does not necessarily represent the official views of the National Institutes of Health. The funders had no role in study design, data collection and analysis, decision to publish, or preparation of the manuscript.

**Competing interests:** The authors have declared that no competing interests exist.

**Abbreviations:** a24pr, anterior area 24 prime; ACC, anterior cingulate cortex; BF, Bayes factor; BOLD, blood oxygen level dependent; cRSN, coarse resting-state network; CV, cross-validation; fMRI, functional magnetic resonance imaging; fRSN, fine resting-state network; GLM, general linear model; MSE, mean squared error; MVPA, multivariate pattern analysis; OP1, opercular area 1/secondary somatosensory cortex; PAG, periaqueductal gray; PCR, principal component regression; PFC, prefrontal cortex; RN, red nucleus; VPM, ventral posterior medial thalamus.

## Introduction

Neuroscience is shaped by a dialectical opposition between modular [1] and global perspectives [2,3], famously underscored by the classic language studies of Broca and Wernicke [4–6]. These showed unequivocal localization of function, but also a distributed character for higher-order processing [4–6]. This dialectic echoes in the present day in discussions of the extent to which a behavior can be reduced to the function of a single brain region or the converse. It plays out in vision [7,8], emotion [9,10], and other areas, but is particularly relevant for pain, where neurosurgical and neuromodulatory interventions are guided directly by scientific understanding of how and where pain experience is localized in the brain [11–13].

Pain is often seen as a distributed phenomenon [14,15]. The distributed model is supported by early functional magnetic resonance imaging (fMRI) data [16–18] and neuroanatomical evidence of widespread nociceptive projections in the brain, including targets in the brainstem and amygdala, [10,19], SII and insula [20,21], and cingulate cortex [22,23]. This has motivated investigations of pain throughout the brain [24] or even as a global brain state [25]. Others have dissected pain from a localist perspective, for example, prioritizing the dorsal posterior insula [26] or anterior midcingulate cortex [27] as "fundamental to" or "selective for" pain. This view follows in the tradition of Wilder Penfield, who first provided support for pain localization through electrical stimulation of dorsal posterior insula [28,29]. Yet, despite 2 centuries of neuroscientific inquiry into the architecture of brain representations, the best way to integrate such perspectives remains the subject of active discussion [26,27,30–33].

Crucially, most brain studies identify foci of evoked pain response [16–18] or multivariate patterns that predict pain intensity [34–36]. Neither of these approaches are sufficient to characterize where pain representations are localized, because they do not explicitly test the spatial scope of representation—whether any region or pattern is necessary or sufficient to decode pain experience. Two pioneering studies advocate for distributed representations by investigating the spatial scope of patterns that discriminate painful from nonpainful stimuli [37,38]. However, they do not comprehensively test either elementary regions or a range of spatial scales. In addition, they do not generalize across diverse task conditions, studies, or types of pain. And, finally, they suffer from a number of statistical limitations, including small sample sizes ($N = 16$), which raises questions regarding their reproducibility and generalizability [39]. Nevertheless, these studies make important contributions, including advancing multivariate modeling as a method of quantifying unique pain information across brain areas.

Here, we characterize the spatial scope of pain representation using multivariate decoding of evoked thermal pain intensity across a diverse set of physical and psychological pain manipulations in a large multistudy cohort ($N = 376$, 11 studies). We formalized a modular perspective by testing decoding models in each of 486 elementary brain parcels (including cortical areas from [40] and others) and in defined resting-state connectivity modules (including 7-network and 32-network scales [41]). We formalized the distributed, multisystem perspective by testing patterns that crossed resting-state modular boundaries, including a priori multisystem maps based on prior literature and the whole brain. Formal model comparisons at each spatial scale quantified the extent to which modular features, distributed networks, and the whole brain represented participants' pain report. Finally, we repeated this process across a selection of datasets with varying data acquisitions and pain manipulations, generalizing our findings to novel studies and contexts.

## Results

### Behavioral data analysis: Pain report is multidimensional

First, we characterized how well our dataset reflects thermal pain experience across diverse task contexts. All but one study (Study 6) varied stimulus intensity and duration (41.1˚C to 50˚C, 1 to 11 seconds), and all but one study (Study 5) had psychological manipulations of pain (pain-predictive expectation cues, Studies 1 to 4; social cues, Study 7; romantic partner hand-holding, Study 6; and perceived control, Study 4). We used linear models of pain report to quantify the cumulative contribution of these factors. Together, they explained 51% of over- all variance in pain across studies (balanced subsample of data, *n* = 112 participants, Fig 1B, S1 Table) and 71% of within-participant pain variance specifically. However, this varied substan- tially between studies: In some cases, psychological factors primarily drove pain variation (Study 2), and in others, sensory factors like thermal stimulus intensity were the most impor- tant (Study 5). This reflects between-study differences in stimulus intensity and duration and in efficacy of psychological manipulations. There were also endogenous nonexperimental fac- tors like sensitization and habituation effects (S1 Table, sens./habit.) or participant-specific idi- osyncrasies (participant mean rating differences), which drove pain reports and, in some cases, dominated (Study 6). Differences in sensitization and habituation were expected due to differences in the number of thermal stimuli and stimulation sites used [42], while systematic differences in pain reports across participants depended on the rating scales (S1 Fig), calibra- tion procedures, and instructions provided in each study. In other words, factors affecting

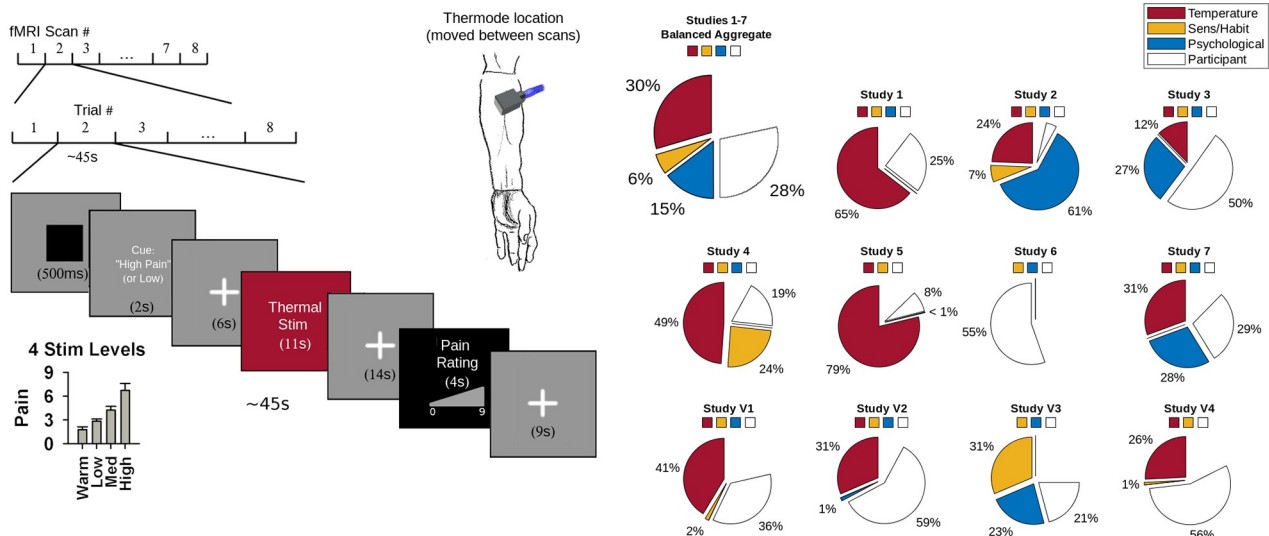

**Fig 1. Study design.** (**A**) Each experimental design involved a succession of thermal boxcar stimuli delivered at variable but finite intervals over a succession of fMRI sessions. A psychological manipulation designed to affect pain intensity preceded the stimulus in many designs, for example, a pain- predictive cue designed to heighten or reduce pain experience. Thermal stimuli varied in intensity and were mostly delivered to the left ventral forearm as illustrated. Cues varied across studies and were uncorrelated with stimulus intensity. After each stimulus, participants rated their pain intensity using a visual scale. For details on each design, refer to Table 2. (**B**) Pain ratings reflect a variety of influences on pain experience. Wedges show the proportion of variance explained by each factor for each study individually and across all studies (excluding validation studies, balancing participants across studies). Since fMRI data were quartiled for computational tractability, and z-scored for between study normalization, these data were also quartiled and (in the aggregated data) z-scored prior to regression analyses. The sum of wedges equals model variance explained. Residual model error is not shown, but completes the circles. We do not consider between-participant differences further in this study, but we display them here to better characterize nonexperimental sources of pain variability. See S1 Table for standardized regression coefficients. Underlying data: https://github.com/ canlab/petre_scope_of_pain_representation/tree/main/figure1. Illustrated arm adapted from Da Vinci's Vitruvian man [101]. fMRI, functional magnetic resonance imaging.

Modeling Spaces

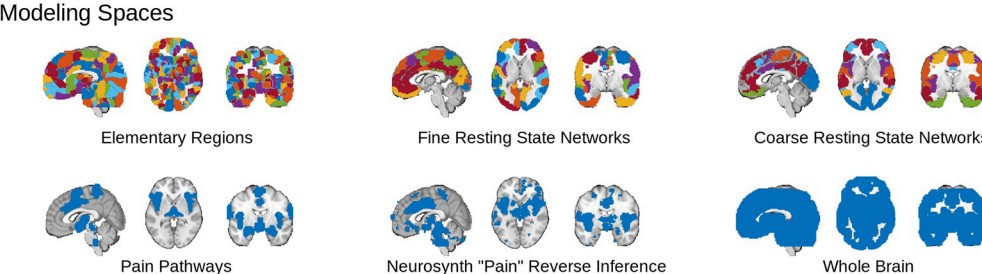

Elementary Regions        Fine Resting State Networks        Coarse Resting State Networks

Pain Pathways        Neurosynth "Pain" Reverse Inference        Whole Brain

**Fig 2. We tested 6 different hypotheses regarding the scope of pain representation by inspecting the decoding performance of models trained on each of 6 corresponding sets of brain areas.** Three modular sets of areas (top row) tested predictive models based on single regions and networks. A single most predictive "module"—region, fRSN, or cRSN—was chosen, and voxel weights within the module were optimized to predict pain. The modules are shown in different colors. Three additional nonmodular sets of areas (bottom row) tested if the BOLD signal across sets of distributed and functionally heterogeneous areas best represented pain. All performance metrics were evaluated using nested CV to ensure independent testing, training, and (where applicable) module selection. Underlying atlas and mask data: https://github.com/canlab/petre_scope_of_pain_representation/tree/main/figure2. BOLD, Blood Oxygen Level Dependent; cRSN, coarse resting-state network; CV, cross-validation; fRSN, fine resting-state network.

pain varied according to variations in study designs and were representative of a swath of the evoked thermal pain experience that spanned both sensory and psychological dimensions.

## Many regions and networks predict pain

Our region and network-level models require selection of a "best" module, i.e. a "best elementary region," best fine resting-state network (fRSN) and best coarse resting-state network (cRSN; "modular" in the lay sense), and there are many candidates (486 elementary regions, 32 fRSN and 7 cRSN, Fig 2, top row; Methods: Defining model spaces and inputs). Module selection warrants special attention because it is a nonlinear step and simulates the selection of regions "fundamental for" pain that has been the focus of recent discussion, so we begin by investigating their individual predictive abilities.

We fit principal component regression (PCR) models to identify optimally predictive patterns within all modules (Methods: MVPA: General method description). Many modules (51/486 elementary regions, 7/32 fRSN and 4/7 cRSN) could decode pain ($p < 0.05$ Šidák-corrected in cross-validated models of predicted versus observed pain with random participant and study slopes and intercepts; Fig 3). No single region stood out as substantially more predictive than all others, with many regional predictions showing heterogeneity across participants and studies, suggesting decisions about the "best" region or network are likely to reflect idiosyncrasies of particular participant samples and experimental paradigms used (median SEM, region: 0.10, fRSN: 0.10, cRSN: 0.10, z-Fisher transformed Pearson $r$ across participants, no random effects, $n = 112$, balanced across studies with $n = 16$ participants per study, SEM illustrated by whiskers in Fig 3). However, some consistent trends were observed.

The areas with the highest predictive accuracy (see Fig 3) included regions implicated in sensory processing and nociception (dorsal posterior insula [OP1], ventral posterior medial thalamus [VPM], and periaqueductal gray [PAG]), motor control (Cerebellar lobule VI and vermis [VII], supplementary motor cortex [Area 6], and red nucleus [RN]) and orienting attention to salient stimuli (lateral prefrontal cortex [PFC; Area 44], frontal operculum [FOP3], and anterior cingulate [Areas 24 and 32]). Among resting-state networks, the "Ventral Attention A" and "Somatomotor" networks were the most strongly associated with pain, although other networks (including "Dorsal attention" and "Visual") were also predictive.

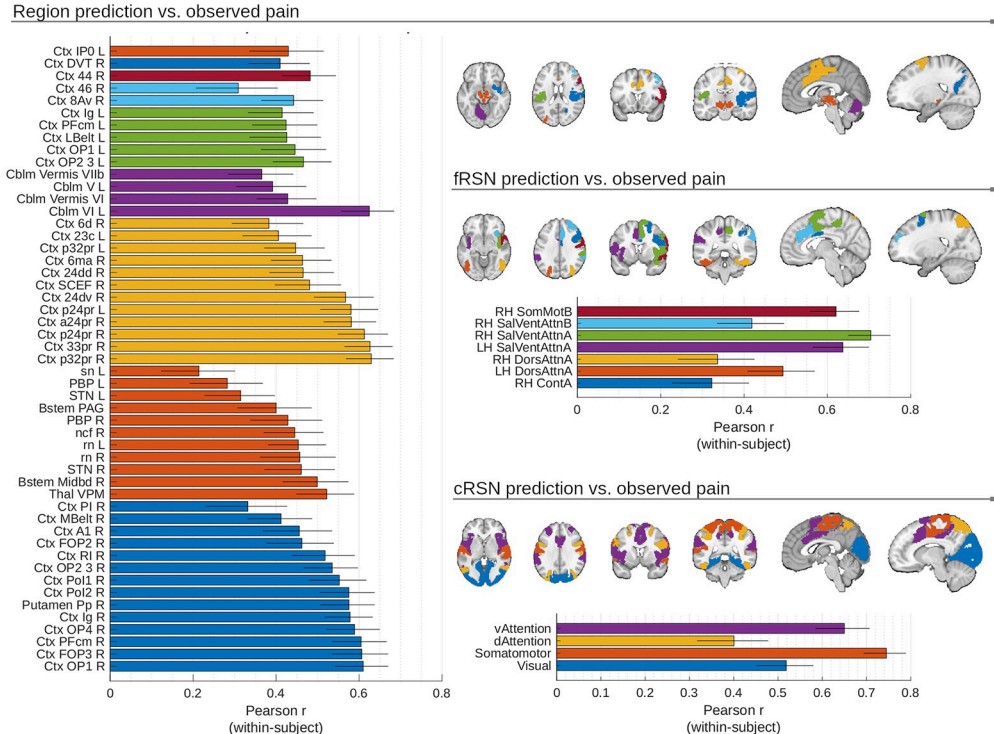

**Fig 3. Multiple brain areas produced models with significant pain decoding capabilities, with prediction accuracy varying considerably across participants (error bars show SEM; *n* = 112).** Predictions from 51 elementary regions, 7 fine-scale (fRSN) and 4 coarse-scale (cRSN) resting-state networks correlated significantly with pain reports (shown; α = 0.05, twice repeated 5-fold cross-validated predictions modeled with random participant and random study effects, Šidák correction for multiple comparisons, effective *p* < 10e-4). This means the "best" region, cRSN or fRSN, is highly dependent on the participant sample examined (and will differ across folds in a CV scheme). Regions are color-coded by large-scale structures (i.e., subdivisions of Ctx, Cblm, Bstem, or Thal). cRSNs and fRSNs are colored by individual network parcels. Underlying data: https://github.com/canlab/petre_scope_of_pain_representation/tree/main/figure3. Cblm, cerebellum; cRSN, coarse resting-state network; Ctx, cortex; CV, cross-validation; fRSN, fine resting-state network; Thal, thalamus.

Additionally, predictive regions were disproportionately in the right hemisphere, reflecting the preferential contralateral representation of thermal stimuli delivered to the left arm (32/51 elementary regions and 5/7 fRSN right lateralized, one elementary region and all cRSN are bilateral; see Table 2 for stimulation sites). These trends in predictive performance across modules provide context for "best" module choices made by our algorithms.

## Multivariate pattern analysis (MVPA) model coefficients are nested across scales

The analysis above describes the distribution of predictive information across the brain at a number of scales, but does not directly show how voxel-level information was used in pain prediction. Multivariate models were fit by first selecting the "best" modules for elementary regions and resting-state networks and then estimating specific patterns of model coefficients (i.e., pattern weights) across voxels for each model (region, network and multisystem/full brain). To further examine how these model coefficients interrelate, we inspected statistically significant coefficients (Supporting information methods: MVPA bootstrap tests) and the similarities in coefficients across spaces. For modular scales, we considered statistically significant

Models share spatial pattern similarities across hypothesis spaces

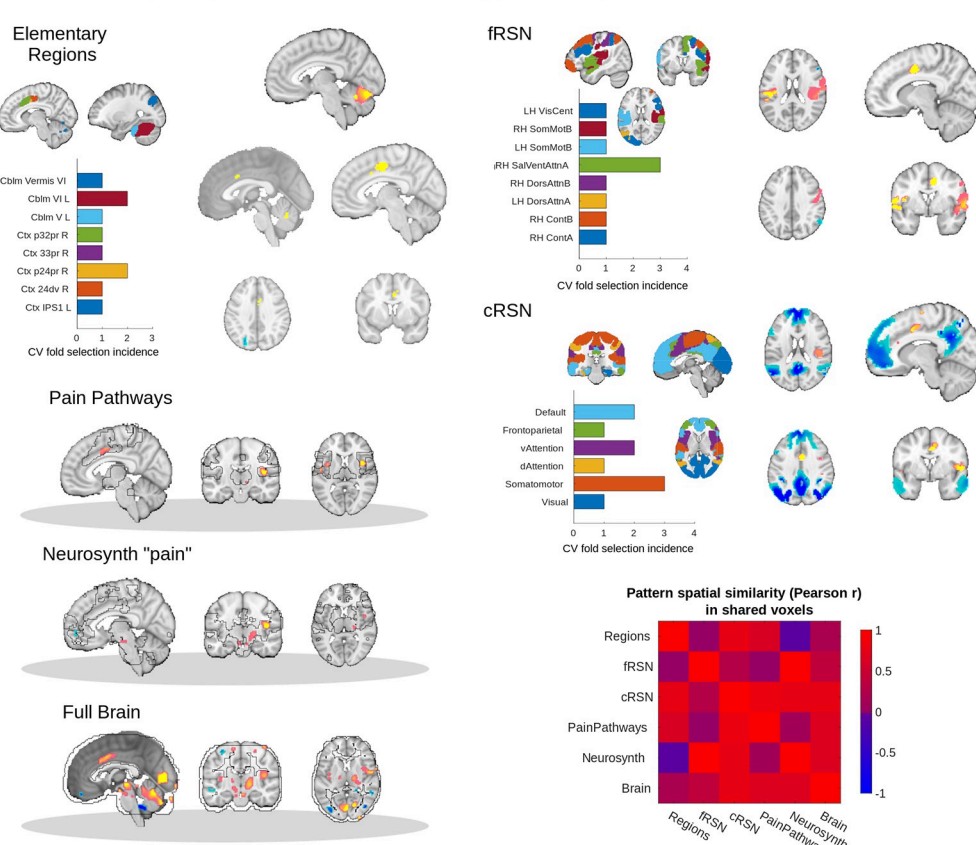

**Fig 4. Multivariate brain models of pain response can be interpreted as nested models that draw on increasing amounts of unique information.** Which module—single-region, fine-scale (fRSN) and coarse-scale (cRSN) resting-state network—was selected by the algorithm varied across folds. Bar graphs illustrate frequency of area selection across 10 folds (2 × 5 repeated CV). Variability across folds is consistent with the large number of highly predictive regions and network parcels and the fact that predictive signal is distributed across them. Bootstrap-corrected maps were computed for all regions to characterize pattern stability. For modular spaces, maps from different CV folds were conjoined after bootstrap correction (i.e., the union of maps across folds), and we calculated pairwise correlations among weights from different models. Model weights are shown on representative brain slices, with yellow/orange indicating positive weights (pro-pain) and blue negative weights (antipain), thresholded with 5,008 bootstrap samples, voxel-wise $p < 0.05$, uncorrected. While the relatively small number of significant weights suggests low model stability, the pattern spatial similarity matrix (correlations in patterns of model weights) shows that these weights were highly consistent across modeling spaces. Across all 6 model spaces, we commonly observed frontal midline deactivations, and cingulate, insular and cerebellar activations, supporting a nested modeling space interpretation. Underlying data: https://github.com/canlab/petre_scope_of_pain_representation/tree/main/figure4. cRSN, coarse resting-state network; CV, cross-validation; fRSN, fine resting-state network.

coefficients for all "best" areas selected during cross-validation (CV; maximum 10 areas, from 2 × 5 CV) and took the union of significant pattern weights for the modules selected across folds, constructing one map for each spatial scale (3 maps, Fig 4).

When fit to the complete dataset (7 studies), the best modules were right anterior cingulate (a24pr), right salience/ventral attention, and bilateral frontoparietal networks, respectively. However, different CV folds chose different "best" modules (Fig 4, barplots). The regions selected appeared to be sampled from areas implicated in sensory processing and orienting responses and were overrepresented in the right hemisphere. Thus, CV predictions from modular spaces reflect the same uncertainty in selecting a single "best" module as we described

above (Fig 3). In the full brain modeling space, significant pattern weights (bootstrap $p < 0.05$) were found in ACC, insular, and visual cortices; PAG; brainstem cuneiform, parabrachial, and raphe nuclei; select cerebellar lobules; and pulvinar, VPM, and ventrolateral nuclei of the thalamus (significant percent voxel coverage in constituent regions relative to permuted null distribution of shuffled voxel indices, $\alpha = 0.05$ uncorrected $p < 0.001$), recapitulating many of the most predictive elementary regions.

To compare model coefficients across scales, we performed a spatial correlation analysis of the intersection of significant pattern weights across pairs of models (bootstrap $p < 0.05$, with modules conjoined across folds where relevant). Pain decoding patterns at one scale were spatially correlated with shared areas of decoding patterns at other scales (mean Pearson $r$: 0.48, interquartile range: [0.11, 0.81], 15 unique pairwise comparisons, Fig 4, correlation matrix). Thus, the voxels with statistically significant weights in the full brain model are largely consistent with the most frequent patterns selected across scales. For example, areas like ACC, insula and cerebellum with significant positive weights in the elementary region, pain pathways, or Neurosynth-derived patterns also had significant positive weights in the full brain pattern. Similarly, negative pattern weights in the ventromedial PFC and dorsal attention areas of the full brain model (significant, $\alpha = 0.05$) were reproduced by patterns fit in analogous areas in the fRSN and cRSN parcels.

This suggests that as we moved from spatially constrained models to spatially less constrained models we in fact built on more constrained models in a consistent way. Larger models exploited the same signals as smaller models and used them in a similar way, but also incorporated additional information available due to the broader scope of a larger modeling space. Thus, each model drew on progressively greater amounts of independent additive information across brain areas as we moved from elementary regions to network parcellations to multisystem and full brain patterns.

## Multistudy MVPA analysis and model generalization performance: Distributed models predict pain better than local models

Comparisons of pain decoding across models at different brain scales ($n = 112$ participants, 16 per study) showed significant differences in performance ($F_{5,11} = 4.25$, $p = 0.021$, mixed model with random participant and study intercepts, random study effects, and Satterthwaite correction for degrees of freedom (df); Fig 5A, S2 and S3 Tables). Models which aggregated across systems performed better. Planned orthogonal comparisons showed models from multisystem (a priori "pain pathways" and Neurosynth-derived) and full brain scales were more predictive on average than modular patterns (regional, fRSN, and cRSN parcellations; $F_{1,6.9} = 15.1$, $p = 0.006$). Otherwise, signatures derived from individual brain modules (regions or networks) performed equivalently to one another, and multisystem models also showed equivalent performance with one another and with the full brain model. Elementary regions were equivalent in performance to cRSN and fRSN (Bayes factor [BF] in favor of the null = 38.9). Neurosynth-based and a priori Pain Pathways-based models were equivalent (BF in favor of the null = 89.2). The full brain model was equivalent to Neurosynth and Pain Pathways models (BF in favor of the null = 15.8). The comparison of cRSN and fRSN did not yield a definitive conclusion ($p > 0.05$ but BF in favor of the null = 5.3, which is less than the $<10$ required for definitive evidence).

One potential limitation is that the statistical contrast of multisystem and full brain decoding versus modular decoding implicitly performed model averaging (average of pain pathways, Neurosynth and full brain on the one hand versus cRSN, fRSN, and elementary regions on the other). This complicates interpretation. For greater transparency and interpretive

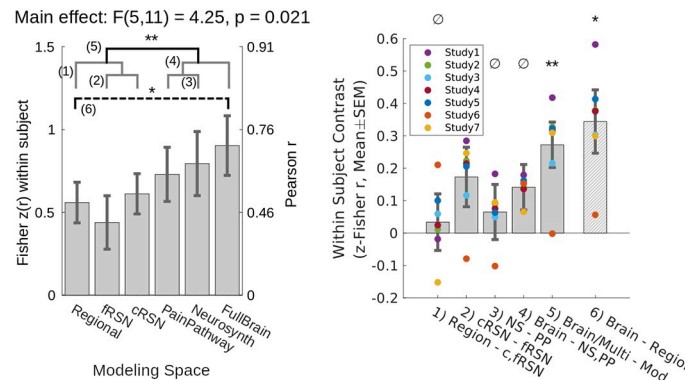

A. Multistudy model performance estimates

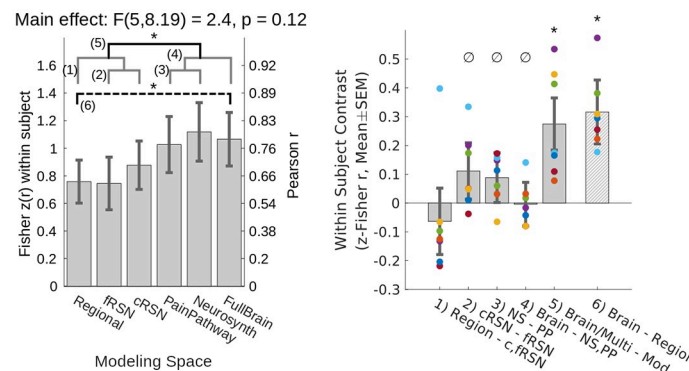

B. Learner performance estimates

**Fig 5. Multisystem and full brain representations of pain performed better than modular representations of pain in a 2 × 5 repeated CV test of model performance.** We Fisher z-transformed within-participant correlation coefficient (predicted versus observed) to quantify performance. **(A)** All obtained models showed significant predictive performance ($p < 0.01$, random participant and study intercept, random study effect), and the specific models obtained here to predict pain using distributed sets of brain areas outperformed the models trained on individual modules (planned comparison, mean(PainPathway, Neurosynth, FullBrain) − mean(regions, fRSN, cRSN) = 0.272 (0.07), $F_{1, 6.9} = 15.1$, $p = 0.0062$). We confirmed the nullity of remaining planned contrasts (BF > 10 favoring the null), except for cRSN > fRSN about which we could reach no conclusion (BF = 5.3 in favor of cRSN = fRSN). Post hoc comparison showed a full brain model outperformed the best single region (FullBrain − Regions = 0.34(0.10), $F_{1,7.9} = 12.39$, $p = 0.008$). Models were trained after pooling data across 7 datasets, yielding 90 participants per training fold. **(B)** We retrained cross-validated models separately for each study to infer whether findings would generalize to new PCR models trained on new datasets. All obtained models showed significant predictive performance ($p < 0.01$, random participant and study intercept, random study effect), but comparison of performance across models from each study showed that distributed representations continued to outperform modular representations (planned comparison, mean(PainPathway, Neurosynth, FullBrain) − mean(regions, fRSN, cRSN) = 0.27(0.09), $F_{1,6.1} = 9.20$, $p = 0.022$). Remaining planned contrasts were null (BF > 10). Post hoc comparison showed training on the full brain results in better performance than training on elementary regions (FullBrain–Regions = 0.32(0.11), $F_{1,5.3} = 8.10$, $p = 0.034$). Training set sizes = {22, 14, 13, 23, 21, 24, 20} participants, resp. All effects estimated with participant-wise repeated measures and random study effects. Mean within study effects overlaid as colored dots. Gray brackets: nonsignificant planned comparisons; black bracket: significant planned comparison; dashed bracket: significant post hoc comparison. Contrasts enumerated across brackets. $^*p < 0.05$, $^{**}p < 0.01$, $^\emptyset$confirmed null. Mixed effect degrees of freedom estimated using Satterthwaite's approximation. SEM error bars. Underlying data: https://github.com/canlab/petre_scope_of_pain_representation/tree/main/figure5. BF, Bayes factor; cRSN, coarse resting-state network; CV, cross-validation; fRSN, fine resting-state network.

convenience, we also performed a planned comparison of the specific difference at the extremes: between full brain decoding (our most distributed model) and the elementary region based decoding (our most local model). The full brain pattern was significantly more accurate ($F_{1,7.9} = 12.4$, $p = 0.008$). This demonstrates the added brain tissue included in the more

distributed models carried unique pain-related signals that were not fully subsumed under more limited subsets of brain areas.

## Learner generalization performance: Distributed multisystem scales produce the best models

Generalization and statistical inference can happen at multiple levels. Above, we generalized the performance of 6 models to a population of participants. However, to quantify information content in an abstract sense, it is important to generalize not only performance, but also learning itself (the model training process). The above analysis did not do this. It lacked multiple independent instances of learning because CV folds shared training data [43]. To generalize over learning instances, we partitioned our data by study, fit models to each study, and then used CV on a study-by-study basis to estimate the performances of these independent models ($n$ = 16 to 30 participants each, 7 studies, 171 participants total; Methods: Study-wise MVPA and learner generalization performance). These model performance estimates varied independently across our sample of 7 studies, which is essential for inference to novel models obtained from novel studies. We refer to this as "learner" generalization performance, borrowing from the machine learning literature.

Regardless of a study's design or experimental manipulation, there was a systematic tendency for models trained at multisystem and full brain scale to outperform models trained on individual modules ($F_{1,6.1}$ = 9.20, $p$ = 0.022, mixed model with random participant and study intercept, random study effects, Satterthwaite df correction; Fig 5B, S2 and S3 Tables). The additional planned contrast between full brain and elementary region models offered a simpler interpretation and confirmed the same result ($F_{1,5.3}$ = 8.10, $p$ = 0.034). This showed that more can be learned in a formal sense by allowing pain representation to span multiple brain systems than by adopting a localized perspective (see S2 and S3 Tables for comparisons in variance explained). For example, the best region approach only explained 81% of the signal represented by the full brain (ratio of $r^2$). Differences between other models were confirmed null (BF > 10) or nearly so (BFs in favor of the null for region = cRSN/fRSN: 9.6, cRSN = fRSN: 37.1, Neurosynth = pain pathways: 75.2, full brain = neurosynth/pain pathways: 98.2).

## Mediation analyses: Models predict multiple dimensions of pain

Our earlier behavioral analysis of pain report showed pain signals originated from multiple sources (Fig 1B), including temperature, expectation, sensitization/habituation, social cues, and perceived control. A good model of pain report should likewise reflect its multifactorial character and mediate the effects of both sensory and psychological influences. Interestingly, behavioral models and our best brain models decoded similar amounts of total pain variance (70% and 72%, respectively), suggesting that brain models might have succeeded in this sense, but we performed a within-participant mediation analysis to formally test the possibility (Supporting information methods: Mediation analysis). Multiple models significantly mediated multiple influences on pain report (S5 and S6 Tables). Here, we focus primarily on the full brain model, which provides a representative summary, because this model subsumes the more spatially constrained models in its pattern weights (Fig 4).

When fit to data from multiple studies, the full brain model significantly mediated the effects of temperature (T), expectation (exp), and habituation (habit) effects on pain report (Fig 6, S4 Table; $T_{ind}$: 0.065 [0.031, 0.113], standardized $\alpha*\beta$ [95% confidence interval], 12% mediated; $exp_{ind}$: 0.030 [0.010, 0.060], 12% mediated; $habit_{ind}$: −0.022 [−0.051, −0.004], 10% mediated). However, the full brain model did not fully capture these effects because direct

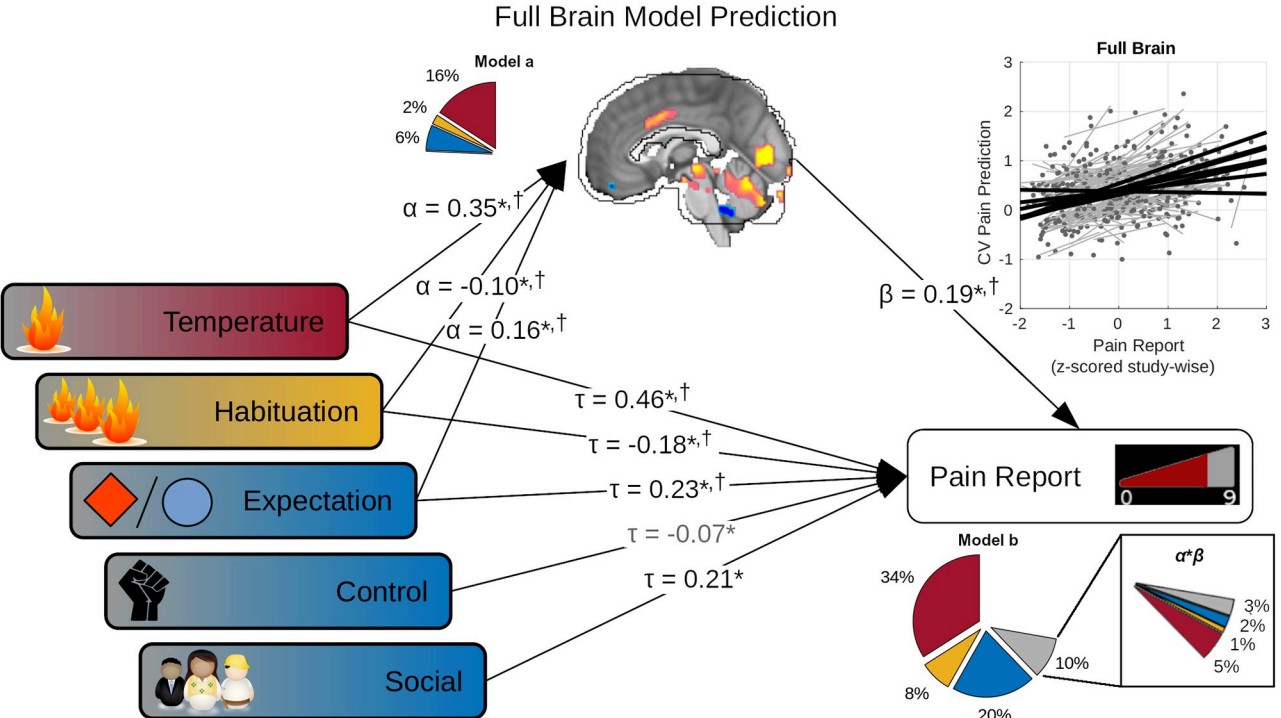

**Fig 6. Full brain models captured brain correlates of both sensory (red, orange) and psychological (blue) influences on pain reports.** Five experimental factors showed significant effects on pain reports: temperature, habituation, perceived control (control), expectation, and social cues (multivariate regression, fixed participant effects, $p < 0.05$). However, the full brain CV model predictions also significantly correlated with pain outcome (scatter plot), and temperature, habituation, and expectation, in turn, significantly correlated with these full brain model predictions ($\alpha$, standardized multivariate regression coefficients; Model a pie chart show partial-$r^2$, color-coded by factor). Finally, the full brain model continued to significantly predict pain outcome ($\beta$) even while controlling for experimental factors ($\tau$). This suggested that temperature, habituation, and expectation effects were captured by the full brain model. Multivariate mediation analysis supported this showing both significant indirect (mediated, $\alpha^*\beta$), and direct ($\tau$) effects on pain report, with additional significant direct effects from perceived control and social cues. Together, these factors and the full brain model predictions explained 72% of pain variance (Model b $r^2$, subdivided by color-coded categories in pie chart; gray: brain model predictions). Of the 10% of variance explained apportioned to the brain predictions in multivariate analysis (model b pie chart), 7% represented indirect temperature, expectation and habituation effects ($\alpha^*\beta$ pie chart insert); however, brain model predictions also explained some unique variance (residual gray wedge in insert). Together, these results showed brain models captured the multifactorial nature of pain experience. Nearly identical conclusions were supported by full brain learner predictions ([†], S4 Table; S5 and S6 Tables for other models). Standardized path coefficients shown. All parameter estimates computed after removing participant fixed effects. [*] $p < 0.05$, in models using full brain model predictions (shown). [†] $p < 0.05$ in models using full brain learner predictions. Gray arrow, significant factor with full brain model but not learner. Bias corrected bootstrap test for nonzero $\alpha$, $\beta$, $\tau$, and $\alpha^*\beta$. Underlying data: https://github.com/canlab/petre_scope_of_pain_representation/tree/main/figure6. CV, cross-validation.

effects remained significant in all cases ($T_{dir}$: 0.471 [0.388, 0.577], partial-$r^2$ = 0.34; $exp_{dir}$: 0.227 [0.148, 0.302], partial-$r^2$ = 0.14; $social_{dir}$: 0.212 [0.177, 0.268], partial-$r^2$ = 0.06; $habituation_{dir}$: −0.139 [−0.211, −0.072], partial-$r^2$ = 0.08; $control_{dir}$: −0.073 [−0.162, −0.001], partial-$r^2$ = 0.01; brain: 0.189 [0.107, 0.276], partial-$r^2$ = 0.1). In addition, the model did not significantly respond to manipulations of social cues or perceived control. Significant mediation of sensory and psychological effects indicated the brain model encoded representations of these effects (brain partial-$r^2_{mediation}$ = 0.07, Fig 6 "$\alpha^*\beta$" insert), but the brain model also decoded additional unique variance in pain reports independent of these manipulations (brain partial-$r^2_{unique}$ = 0.03, "model b" total brain partial-$r^2$ = 0.10). This showed our full brain model captured multiple sources of influence on pain report, in proportions representative of their overall effect on experimental pain, but missed the effects of some variables (social cue and perceived control), while also decoding variance in pain unrelated to the experimental manipulations. Mediation effects at other scales generally captured some subset of the above (S5 Table), but expectation effects were only significantly mediated by full brain, Neurosynth, and cRSN models.

While this mediation analysis showed some examples of mediation, it does not generalize to new models which might be learned in new studies. The latter requires mediation of model learner performance, i.e., performance of models trained independently across studies. However, mediation analysis of study specific models did not show any significant mediation effects (Full brain models: Study 7 social mediation $p < 0.1$, Study 2 temperature mediation $p < 0.1$, all others $p > 0.1$), likely due to a known lack of power in bootstrap tests of small samples [44]. We therefore pooled predictions obtained from each study's models and performed a within-participant mediation analysis across studies for each learner (S4 and S6 Tables). For full brain learners, temperature, expectation, and habituation showed significant partial mediation ($T_{ind}$: 0.129 [0.088, 0.175], 23% mediated; $exp_{ind}$: 0.028 [0.005, 0.056], 13% mediated, $habit_{ind}$: −0.020 [−0.041, −0.002], 12% mediated), while temperature, expectation, social cues, and habituation showed significant direct effects on pain report ($T_{dir}$: 0.429 [0.353, 0.510], partial-$r^2$ = 0.31; $exp_{dir}$: 0.192 [0.122, 0.259], partial-$r^2$ = 0.10; $social_{dir}$: 0.221 [0.188, 0.251], partial-$r^2$ = 0.06; $habit_{dir}$: −0.149 [−0.207, −0.092], partial-$r^2$ = 0.06) while controlling for brain predictions (brain: 0.253 [0.186, 0.315], partial-$r^2$: 0.16, partial-$r^2_{mediation}$: 0.12, partial-$r^2_{unique}$: 0.04). These results mirrored those found in the multistudy analysis (Fig 6, S4 Table), albeit with perhaps a greater bias toward capturing sensory rather than psychological effects. These effects were once again largely consistent across scales, with a greater tendency for distributed learners to mediate psychological effects (expectation and social cues; S6 Table).

It was not possible to understand which specific experimental factors were mediated by multivariate learners in any particular study, nor was it possible to distinguish whether learners captured multiple factors simultaneously or if models obtained from different studies rather differed in the factors they mediate. Nevertheless, these results do show that representations of multiple distinct influences on pain experience were learned, especially at multisystem and full brain scales. This supports the notion that distinct sensory and cognitive pain modulating signals were captured by multisystem spaces in a general sense.

## Validation

We validated both our models' and learners' performances by evaluating 4 studies, which once again differed in experimental design, but which we did not use for model training or obtaining estimates of model or learner performances (Methods: Data). This included inspection of statistical model calibration, i.e., whether observed validation data fell within the confidence intervals of initial performance estimates, given the precision of those initial performance estimates (Fig 7, prediction 95% confidence intervals indicated by violin plots). Variability in decoding accuracy across validation studies was consistent with our initial estimates, suggesting that our inferences were well calibrated (Fig 7A, observed: connected dots). We also inspected significant comparisons from the model development stage in the validation studies and found that validation data fell within the range of expected effect sizes for multisystem and full brain versus modular comparisons but not brain versus elementary region comparisons.

Multisystem and full brain models showed significantly better accuracy than modular region or network models ($F_{1,1005}$ = 15.6, $p$ = 8.1e-5, mixed model with fixed study effects, random participant intercept, Satterthwaite df correction). At the model (as opposed to learner) level, full brain decoding was not significantly better than elementary region decoding in the validation data ($p$ = 0.68, BF = 3.3e4 in favor of full brain = regions). However, in tests of learner generalization performance, multisystem models significantly outperformed modular region or network models (multisystem and full brain versus region, cRSN, fRSN: $F_{1,1005}$ = 25.3, $p$ = 5.7e-7; Fig 7B), the full brain models outperformed single-region models (brain versus region: $F_{1,201}$ = 6.5, $p$ = 0.01), and learner performance fell within the expected range for

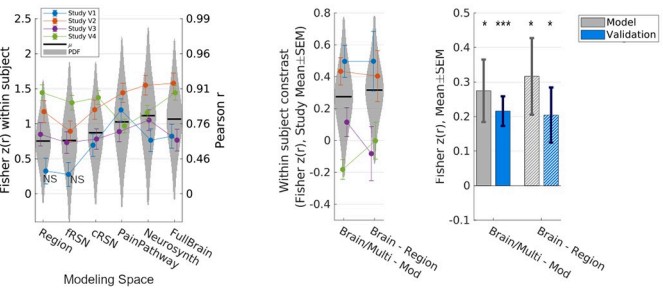

**Fig 7. Distributed representations outperformed modular representations in validation datasets, consistent with estimated generalization performances. (A)** All models fit to the 7 study dataset significantly predicted pain in novel data, except for 5 models in 5/28 cases ($p < 0.05$, left, only nonsignificant results labeled, Holm–Sidak corrected). Performance in validation data continued to be significantly better for multisystem and full models relative to modular models in validation studies 1 and 2 (mixed effects model, random participant intercept, $F_{1,160} = 12.6$, $p < 10e-3$; $F_{1,220} = 11.8$, $p < 10e-3$; resp.) as well as overall (mixed effects model, random participant intercept, and fixed study effects, $F_{1,1005} = 15.6$, $p < 1e-4$); however, the full brain model did not show better performance than the best elementary region model specifically (region = full brain, BF = 2.2e5; right). **(B)** Nevertheless, retraining all 6 families of algorithms in novel data reproduced both findings: full brain and multisystem representations predicted more pain than regional and network representations and the full brain model specifically out performed any single best region. In a $2 \times 5$ repeated CV scheme, all models separately retrained for each study successfully predicted pain, except for 2/28, specifically in validation Study 1 (left, only nonsignificant results labeled, Holm–Sidak corrected); however, full brain and multisystem models significantly outperformed modular models in validation Studies 1 and 2 (mixed effects and random participant intercept, $F_{1,160} = 15.0$, $p = 0.0002$; $F_{1,220} = 28.4$, $p < 1e-6$; resp.) and overall (mixed effects, random participant intercept, and fixed study effects, $F_{1,1005} = 25.3$, $p < 1e-6$). Additionally, the full brain models specifically outperformed the best elementary region models in Study 2 (mixed effects, random participant intercept, $F_{1,44} = 10.1$, $p = 0.0027$) and overall (mixed effects, random participant intercept, and fixed study effects, $F_{1,201} = 6.49$, $p = 0.012$; right). $^{NS}p > 0.05$ (not significant), $^*p < 0.05$, $^{**}p < 0.01$, $^{***}p < 0.001$, $^{\emptyset}$BF > 10, "strong" evidence of null. Gray bars previously shown in Fig 5. Error bars: SEM, random participant intercepts, random study (gray bars) or fixed study (otherwise). Violin plot width indicates likelihoods estimated by inferential models fit to the original dataset (7 studies; Fig 5), vertically constrained to 95% confidence intervals. Underlying data and validated models: https://github.com/canlab/petre_scope_of_pain_representation/tree/main/figure7. BF, Bayes factor; CV, cross-validation.

both (Fig 7B). These comparisons confirmed the validity of inferences drawn: In novel participants and experimental designs, decoding at multisystem and full brain scale systematically outperforms more spatially constrained and targeted models of pain intensity, supporting the hypothesis that pain information is broadly distributed across multiple brain systems.

## Models capture pain-specific representations

We next investigated the pain specificity of learned models using an independent multimodal dataset of within-participant contrast maps (task versus baseline averaged across trials, or mean high versus low magnitude stimulus evoked blood oxygen level dependent, BOLD,

Models discriminate pain from putative homomorphs

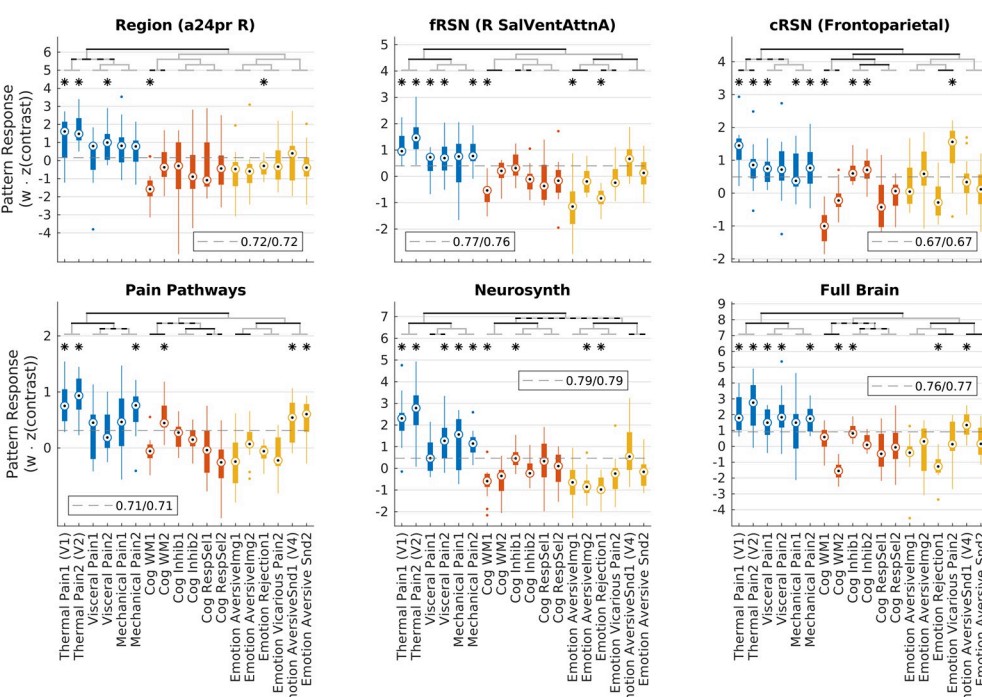

**Fig 8. Distributed models were more pain-specific than local models.** In a set of 18 studies that estimated participant mean responses to 1 of 6 multimodal pain tasks or 1 of 12 nonpain tasks, models significantly discriminated pain from nonpain tasks on average ($p < 10e-6$), even though nonpain tasks all shared aversive characteristics or task demands with pain. Planned contrasts showed this discrimination was statistically significant for each of the 6 models tested. Notably, all models were exclusively trained on painful stimuli, so the ability to discriminate was an emergent phenomenon. Most importantly, discrimination of pain from nonpain was greater for distributed than local models ($p = 2.2e-5$), indicating that the performance advantage of distributed models was due to improved pain-specific representations. Bars are grouped into pain tasks (blue), cognitive demand control tasks (red), and nonpainful aversive tasks (yellow). Planned contrasts: black brackets, $p < 0.003$ (Sidak threshold for 17 comparisons), gray/black brackets $p < 0.05$. Post hoc $t$ test: $^*p < 0.003$. Dashed gray line: optimal cut point for discrimination (balancing sensitivity and specificity). Legend indicates sensitivity/specificity at the optimal threshold. Underlying data: https://github.com/canlab/petre_scope_of_pain_representation/tree/main/figure8. cRSN, coarse resting-state network; fRSN, fine resting-state network.

signal) from 18 different studies. These include both thermal, pressure, and visceral (rectal distention) pain (6 studies, 2 per modality, $n = 15$ participants per study) and studies that shared negative affective dimensions or task demands with pain rating. The latter consisted of tasks focusing on aversive images, social manipulations (romantic rejection and vicarious pain), and aversive sounds, and tasks focused on working memory, response inhibition, and response selection (2 studies × 15 participants each). Specificity was tested by comparing model scores for multimodal pain studies to those from the cognitive and affective studies with putatively homomorphic evoked responses. To determine whether models were specific to heat pain or specific to more general pain representations, we also tested generalizability by comparing model scores between heat pain versus pressure and visceral pain contrasts.

On average models produced greater scores when applied to mechanical, thermal, or visceral pain data than when applied to putatively homomorphic tasks like response inhibition, aversive sounds, or emotional rejection (Fig 8; B = 1.19, $t_{270} = 18.5$, $p < 10e-6$, planned contrast, mixed effects model with random participant effects), and although not trained as

classifiers our models nevertheless discriminated pain from putatively homomorphic tasks (sensitivity/specificity for pain versus not pain at optimal cut point, right a24pr (elementary region): 0.72/0.72, right salience/ventral attention (fRSN): 0.77/0.76, frontoparietal (cRSN): 0.67/0.67, pain pathways: 0.71/0.71, neurosynth: 0.79/0.79, full brain: 0.76/0.77). These models were all trained on heat pain stimuli. Consequently, in addition to preferentially responding to pain versus putative homomorphs, all models responded preferentially to heat pain tasks over mechanical or visceral pain (B = 0.75, $t_{270}$ = 6.7, $p$ < 10e-6, planned contrast), but most models also showed significant predictions in most visceral and mechanical pain studies (fRSN, cRSN, neurosynth and full brain models predicted 3 out of 4 visceral and mechanical studies, $t$ test, $p$ < 0.05, corrected for 18 comparisons, Fig 8). Most importantly, discrimination of pain from not pain tasks was significantly better for larger scale models (distributed − local = 0.34, $t_{1350}$ = 4.3, $p$ = 2.2e-5, planned contrast). This advantage could not be attributed to greater specificity to the primitive stimulus features of our training data because the difference between heat and visceral/mechanical pain was unchanged across models (BF = 21.2): the move from local to distributed representations increased pain identifiability equally across all 3 pain modalities, without a preference for heat pain. Thus, our models discriminated pain from not-pain, larger scale representations discriminated better, and this effect was due to their ability to capture a better multimodal representation of pain rather than any increased specificity to the particular task characteristics of our training data. These results confirm that the models learned in this study captured pain-specific information.

## Discussion

This study investigated the spatial scope of pain representation. In a sample of participants pooled across 7 studies, we found that evoked pain intensity was best decoded when pooling information from multiple brain systems. However, models trained on the full brain showed no advantage over multisystem models, suggesting the latter had already subsumed all uniquely informative signals. Meanwhile, an elementary region account emerged as a surprisingly good approximation of overall pain representation. Follow-up comparisons of models trained separately for each study support identical conclusions, showing that these results will generalize to new models which might be learned from novel participants, experimental designs and studies. These results build on prior studies that examined only one particular spatial scale a priori (e.g., [34,36]) or investigated spatial scope of pain-predictive representations under more restrictive conditions [37,38] by comprehensively quantifying information representation across scales and generalizing to heterogeneous evoked pain experiences across a diverse collection of task conditions, including mechanical and visceral pain. We thus demonstrate that evoked pain is represented preferentially by specific brain areas, but that multiple areas are required for accurate pain decoding.

Several limitations of our approach must be recognized. This study assumes that there is a common configuration of brain activity that corresponds to pain representation, that it can be measured using BOLD fMRI, and that incremental reconfigurations of this activity correspond to proportional changes in subjective experience. However, distinct representations have been demonstrated for different types of pain including somatosensory and vicarious pain [45] and different kinds of clinical pain [46,47]. Further, BOLD fMRI is best at measuring the anatomical configuration of responses [48], but the absolute magnitude of evoked activity also encodes pain information [26]. Finally, structural and functional heterogeneity across individuals as well as physical limitations on spatial resolution erodes local information decoding capabilities of fMRI, while the complexity of distributed representations makes them much harder to identify with multivariate methods. Nevertheless, the specificity and consistency of our findings

across heterogenous studies and task conditions supports the notion that good pain representations were successfully obtained here.

Multivariate fMRI models formally map specific configurations of brain activity to constructs of interest, but constructs must be validated [49]. Here, the performance of multivariate representations was reproducible across constituent studies, and all models showed significantly greater scores during pain tasks than superficially correlated cognitive or affective tasks. Differences ranged from moderate to dramatic: equivalent to a diagnostic odds ratio of 14:1 for the Neurosynth-derived model. Furthermore, all models generalized to other pain modalities like mechanical and visceral pain that were also distinguished from nonpain tasks, and this multimodal specificity tracked heat pain specificity across models. Some models idiosyncratically tracked some nonpain modalities like the odd putatively homomorphic task. This is unsurprising because our models were designed to capture all pain relevant features, including any shared with other tasks, mediation analysis suggested they captured some (e.g., anticipation), and linear models cannot discriminate nonspecific features based on context. Although we cannot extend our conclusions to characterize all forms of pain, these results show both convergent and discriminant construct validity for multimodal evoked pain [49]. Therefore, multivariate modeling captures good approximations of common, cross-subject evoked pain representations.

Despite the strengths of multivariate brain models, this study moves beyond the performance of individual models to make stronger claims about information content. Our multistudy analysis provides a convenient and familiar starting point of fitting a model to data and establishes our ability to estimate model performances accurately. However, unlike traditional unbiased statistical models, machine learning is not optimized to recover underlying generative processes [50,51]. Similarly, the routine exercise of generalizing from one statistical model's performance to new models that might be learned is notoriously problematic in machine learning [43]. We resolve these issues by operationalizing "brain information content" as what can be learned in general and simply estimate performance of independent models retrained for each study separately (i.e., 42 different models, 6 hypotheses, and 7 studies). It is conceptually simple, follows precedent established in machine learning [52,53], applications in genomics [54], and more general modeling recommendations for multistudy mega-analyses [55]. The consistency of our results across heterogenous evoked heat pain tasks bolsters construct validity, but such qualitative implications are incidental. More importantly, evaluating variability of independent model performances across studies using traditional statistical methods quantitatively informs the performance of new unseen models trained on unseen new participants in new studies in a practical sense. We successfully validated these estimates and illustrated the broader nature of this claim by training new models in new studies on new participants and obtained an identical performance hierarchy favoring multisystem representations. This provides a blueprint for exploiting large heterogeneous datasets to practically quantify information content in the brain, one which can just as readily be applied to other brain phenomena besides pain.

Although our findings primarily support a multisystem representation of pain, they also support the notion that specific regions have a privileged role in pain representation. Different subregions of mostly ACC, insula, midbrain, and cerebellum significantly predicted pain on their own. Previously, some of these areas have been suggested to be pain specific (a24pr) [27] or "fundamental" to pain (OP1) [26] (regions labeled using the scheme of [40]). Substantial evidence indicates cingulate and insular cortex represent parallel pathways mediating distinct dimensions of pain. Cingulate cortex has traditionally been associated with emotional–motivational dimensions of pain [23,56–61]. Conversely, posterior insula has traditionally been associated with sensory-discriminative dimensions [21,29,62]. Convergence between our results

and prior studies puts such studies in a quantified context. We showed that many brain regions might predict pain with high sensitivity depending on the experimental conditions and sample of participants observed, but there was substantial unique information elsewhere in the brain (24% more in the full brain versus best elementary region), which regional interpretations of pain failed to capture.

Resting-state network modules also predicted pain intensity, especially somatomotor, salience, and attention networks. Salience networks respond to painful stimuli [63–66], leading to interpretations of pain as an orienting response [14]. Our results echo network level interpretations of pain, but brain network parcels of pain were no more pain predictive than elementary regions, correlated with regional pattern maps, and the salience and somatomotor regions selected most frequently showed substantial overlap with individually predictive insular and cingulate elementary regions. This indicates that the best a network model of pain can hope for is to recapitulate the information already present in the best constituent elementary region.

Multisystem and full brain representations were the most pain predictive, most specific, and supported the greatest learning capacity across studies. Converging evidence implicates anterolateral, midbrain, and cerebellar brain systems in cognitive evaluation, more posterolateral areas in detection of physical stimulus properties, and thalamic and frontal midline regions in both cognitive and physical evaluation of painful stimuli [60,67–77]. Although characterizing the diversity of pain information in the brain exceeds the scope of this study, this was precisely the scope of [78] and [79] who use Study 2 and Study 5 (respectively) to identify expectation mediators in dorsolateral PFC, amygdala, pons, and dorsal striatum, thermal stimulus mediators in somatomotor areas and cerebellum, and mediators of both components in dorsal ACC, anterior insula, and thalamus. These results are all consistent with statistically significant weights in our multisystem and full brain models and with our mediation analysis, which showed mediation of expectation and stimulus intensity effects. This lends support to the notion that multisystem models subsume, integrate, and expand upon a diverse constellation of unique regional signals.

Several studies suggest that traditional methods of functional localization may miss subtle representations within and across brain areas and that pain representations might be embedded in areas typically associated with unrelated processes [8,24]. We hoped the full brain space would allow our models to capture traditionally overlooked signals, but the failure of full brain models to outperform multisystem models suggests that any such exotic signals are redundant with what was already captured by the multisystem representations.

We used MVPA models of pain defined at spatial scales according to established biological gradients in anatomy and function, or a meta-analytic area representative of the neuroimaging findings of the field, estimated their performance, and validated these estimates to show multisystem models best represent pain. We then generalized our conclusions to novel models trained in novel studies and validated these conclusions as well. Finally, we showed our models captured pain in a multidimensional manner and were specific to multimodal evoked pain but not other experiences with common cognitive characteristics, affirming construct validity. In the face of current gyrations between local and global perspectives of pain, our results illustrate the extent to which the neural basis of pain can be localized, that it is best captured by distributed signals spanning multiple brain systems, and provide a blueprint for quantifying information content in the brain across diverse neural phenomena.

## Methods

### Participants

This study aggregated data from 11 previously published studies involving 376 participants from whom written consent for participation was obtained. Institutional review boards at the

**Table 1. Study participants overview.**

| | Sample size | Gender | Mean age (SD) | Citations |
|---|---|---|---|---|
| Study 1 (BMRK4) | 28 | 10F | 25.7 (7.4) | [45] |
| Study 2 (EXP) | 17 | 9F* | 25.5 | [78] |
| Study 3 (IE2) | 16 | 9F | 25.9 (10.0) | [95] |
| Study 4 (ILCP) | 29 | 16F* | 20.4 (3.3)** | [96] |
| Study 5 (NSF) | 26 | 9F | 27.8 | [34,79] |
| Study 6 (Romantic pain) | 30 | 30F | 24.5 (6.7) | [97] |
| Study 7 (SCEBL) | 25 | 11F | 27.4 (8.8) | [98] |
| Study V1 (BMRK3) | 33 | 22F | 27.9 (9.0) | [34,75] |
| Study V2 (IE) | 45 | 25F | 24.7 (7.1) | [99] |
| Study V3 (Placebo) | 40 | 0F | 26 [19,40]*** | [100] |
| Study V4 (BMRK5) | 87 | 46F | 28.5 (5.7) | [82] |

*Gender of one participant unknown.

**Age of one participant unknown.

***Standard deviation unavailable; range provided instead.

University of Colorado Boulder, Columbia University, or the ethics committee of the Medical Chamber Hamburg approved these studies. Please refer to Table 1 for details on sample size, age ranges, and gender. All participants were right handed.

## Data

Except for Study V4, data were included in this study if they conformed to several criteria: (1) they included evoked pain stimuli and subjective ratings of the intensity of those stimuli; (2) they were previously published at the time this study was conceived (May 2019); (3) involved healthy participants, without any drug manipulations (excluded one study [74]); (4) if the lead authors on those prior publications had prepared a set of single-trial stimulus evoked contrast images (see "General linear model (GLM) analysis" subsection below) in the course of their original analyses for purposes unrelated to this study; and (5) if those contrast images had been made available to the senior author of this study for archival purposes (TW). The latter requirements introduced organic variability across datasets more representative of variability seen across studies in the field. Sources of variability include differences in preprocessing choices and statistical designs. Study V4 was introduced subsequently to provide multimodal data (pain and aversive sound) for improved tests of model specificities, but otherwise conforms to the same requirements. Refer to Table 1 for references to publications containing relevant experimental design and data acquisition details and Table 2 for overviews of experimental designs. Data are available as "single trial" statistical contrast maps, hosted at figshare.com (uniformly prepared by BP), and best accessed via the CANlab Single Trials Repository on GitHub (https://github.com/canlab/canlab_single_trials).

Each study involved a series of trials of painful thermal stimuli of varying intensities (40.8 to 50°C) applied to the left arm, anticipatory cues, noxious stimuli lasting 1 to 20 seconds, and a pain rating period following a variable (jittered) interval (Fig 1A). Additionally, many studies included cognitive or affective psychological manipulations like romantic partner hand-holding or manipulations of perceived control over stimulus intensities (Fig 1B). Table 2 details stimulus durations, sites, intensities, and additional protocol details by study. S1 Table provides effect sizes for experimental pain manipulations across studies. S1 Fig shows histograms of pain ratings across studies.

**Table 2. Study stimuli overview.**

|  | Stimulus intensity (°C) | Stimulus duration | Stimulus location | Thermode size (mm²) | Trials per participant | Other experimental manipulations |
|---|---|---|---|---|---|---|
| Study 1 (BMRK4) | 46, 47, 48 | 11 seconds | L forearm, L foot | 16 | 81 | Learned heat-predictive visual cues |
| Study 2 (EXP) | 41.1, 44.2, 47.1* | 10 seconds | L forearm | 16 | 64 | Learned heat-predictive auditory cues |
| Study 3 (IE2) | 48, 49 | 1 second | L forearm | 27 | 70 | Learned heat-predictive visual cues |
| Study 4 (ILCP) | 45, 47* | 10 seconds | L forearm | 16 | 64 | Perceived control, learned heat-predictive visual cues |
| Study 5 (NSF) | 40.8 to 47.0 | 11 seconds | L forearm | 16 | 48 | Masked emotional faces |
| Study 6 (Romantic pain) | 47 | 11 seconds | L forearm | 16 | 16 | Romantic partner hand-holding |
| Study 7 (SCEBL) | 48, 49, 50 | 2 seconds | R calf | 27 | 96 | Unlearned visual expectation cue |
| Study V1 (BMRK3) | 44.3, 45.3, 46.3, 47.3, 48.3, 49.3 | 12.5 seconds | L forearm | 16 | 97 | Cognitive self-regulation of pain (reappraisal) |
| Study V2 (IE) | 46, 47, 48 | 4 seconds | L forearm | 16 | 48 | Learned heat-predictive visual cues, placebo |
| Study V3 (Placebo) | 46.4* | 20 seconds | L forearm | 30 | 60 | Unlearned visual cues placebo |
| Study V4 (BMRK5) | 47, 48, 49 | 8 seconds/11 seconds | L forearm | 16 | 36 | None |

*Variable across participants. Mean across participants shown.

We performed all of our primary analyses in 7 studies (Studies 1 to 7) and used 4 validation studies (Studies V1 to V4) as a post hoc confirmation of the validity of our inferences. Datasets were selected as validation studies semirandomly. We aim for our results to generalize to novel studies and contexts; however, we recognize that we cannot exhaustively include all possible permutations of studies and contexts in our analysis here. To simulate the burden of generalizing to novel pain conditions, we selected validation studies that partially overlapped with the experimental conditions of the training dataset (studies V1, V2, and V3 had thermal intensity variation and V2 had expectation cues) as well as studies that had novel experimental conditions absent from the primary analysis (study V1 had pain reappraisal and V2 and V3 had placebo manipulations). Additionally, we included studies with multimodal stimuli (V1 and V4) so that we might evaluate pain specificity of decoding models (S1 Text). Importantly, the use of validation studies was separate from the use of CV, which was used extensively for model fitting and evaluation (see MVPA sections below): CV generates an estimate while validation studies put that estimate to the test. Analysts were blind to validation studies until all primary analyses had been completed.

In addition to the "single trial" dataset used to test pain intensity decoding, we also assembled a partially (but minimally) overlapping dataset of traditional GLM contrast maps (one contrast per participant representing a mean task effect) from multimodal pain and nonpain studies for indirect evaluation of model pain specificity. This dataset consisted of 18 previously published studies: 2× acute thermal pain tasks (Study V1 and V2 above), 2× visceral pain tasks [80,81], 2× mechanical pain tasks [82], 2× working memory tasks (WM) [83,84], 2× response selection (Inhib) [85,86], 2× response conflict (RespInhib) [87, 88], 2× negative emotion induction through visual scenes (AversiveImg) [89,90], 1× social rejection (Rejection)[91], 1× vicarious pain (Vicarious Pain) [45], and 2× emotionally aversive vignettes from the International Affective Digital Sounds system (Emotion AversiveSnd, including study V4 above) [88]. Each study contributed 15 contrast maps from 15 participants, and the dataset is designed to balance across studies and task categories, which consist of pain tasks (6×), cognitive demand

control tasks (e.g., pain rating requires holding pain experience in working memory, inhibition of defensive withdrawal reflexes, etc.; 6×) and tasks that share aversive emotional qualities with pain (6×). This dataset is a modified version of a previously published dataset [88] available at https://neurovault.org/api/collections/3324. Contrasts of stimulus versus baseline from Study V1 and Study V2 were substituted for the original thermal pain studies, since coincidentally the original thermal pain studies from [88] are used to train our decoding models here (Study 2 and Study 5 in particular), and we wished to test models on independent data.

## Behavioral data analysis

To provide an informative description of each study's experimental effects, we modeled pain reports as a function of experimental manipulations while controlling for participant fixed effects (differences in mean pain between participants). We considered all experimental manipulations, as well as site specific and nonspecific sensitization and habituation effects and then used backward stepwise regression to select parameters which affect pain report ($p < 0.05$). All nonsignificant parameters other than mean participant effects were removed. For simplicity, this analysis did not consider nonlinear effects (i.e., interaction effects). All analyses were performed in quartiled data to match and thus better inform the subsequent MVPA analyses, which were similarly quartiled due to computational constraints (see Data standardization subsection below for details).

## Defining model spaces and inputs

We defined 3 hypotheses regarding scope of pain representation (modular, a priori multisystem areas or full brain) and formalized them using one or more of 6 modeling spaces (explicitly enumerated below), which constrained subsequently trained machine learning algorithms to draw on sources of information from particular brain areas (Fig 2).

The "modular" hypothesis contends that pain intensity information is best captured in a single physiologically coherent brain area, but is agnostic with respect to location and size of the brain area. To formalize this, we drew on 3 brain parcellations that vary in areal size from fine to coarse. (1) Our finest representation, the "elementary region" parcellation, defined a set of elementary brain areas by combining the Human Connectome Project's cortical parcellation [40] with individual cerebellar lobules, and thalamic basal ganglia and brainstem nuclear parcellations. (2) A fRSN parcellation involving 32 distinct lateralized resting-state network parcels and (3) a cRSN parcellation using 7 bilateral parcels formalize the network hypothesis. Both fRSN and cRSN parcellations were derived from clustering of brain areas based on their functional connectivity profiles, but differ in their clustering threshold. Illustrations (Fig 2) show that some networks primarily involve adjacent areas while others involve several disconnected areas of the cortex.

The hypothesis that pain has a "multisystem" representation assumes that there is a pain-specific configuration of brain activity localized in a finite but not necessarily contiguous or physiologically homogenous set of brain areas. We formalized this hypothesis as 2 additional modeling spaces. (4) First, by identifying a priori the subset of the regional parcellation that receives afferent nociceptive information either directly or indirectly and taking its union to define a model space ("pain pathways"). These regions include thalamic and brainstem nuclei, amygdala, insular, somatosensory, and mid-cingulate cortex. (5) Next, we took an empirical approach using a reverse inference metaanalysis map for "pain" from neurosynth.org. Both pain pathways and neurosynth spaces crossed multiple modular boundaries and comprised approximately 30% of the brain, including cortex, thalamus, subcortical nuclei, and brainstem regions.

Finally, the holistic hypothesis assumes that any of a number of sensory or cognitive functions may recruit any one brain area, and, therefore, pain-predictive information could be distributed anywhere and everywhere in the brain. (6) Models trained on the entire brain formalized this final hypothesis.

Thus, 3 model spaces formalized the modular hypothesis, 2 formalized the multisystem hypothesis, and 1 formalized the holistic hypothesis. We elaborate on how each space was defined in the Defining mode spaces and inputs in S2 Text.

### fMRI acquisition and preprocessing

Briefly, all data were acquired using 3T MRI scanners and were motion corrected, had cerebrospinal fluid and white matter signals linearly removed, and were spatially normalized to a standard MNI152 template, except for Studies 2, 5 and V4. Studies 2 and 5 were acquired at 1.5T, and none of the 3 had cerebrospinal fluid or white matter signals removed. Please refer to studies referenced in Table 1 for further details on data preprocessing.

### General linear model (GLM) analysis

This study used beta-series analysis [92] to estimate brain response to each evoked noxious stimulus (a "single trial"). The GLM design matrix included separate regressors for each stimulus event and nuisance parameters (e.g., head motion). This study excluded trials with variance inflation factors >2.5. We thus obtained a separate parameter estimate brain map for stimulus evoked activity for each stimulation (trial), which represents heat versus baseline. For computational tractability, we averaged these maps within pain intensity quartile after data standardization (see "Data standardization" below for discussion).

### Data standardization

We demeaned data across studies (subtracted the study mean map from all constituent contrast maps) and standardized all single trial images (mean voxel value is 0, and standard deviation of voxel values within trial image is 1). Thus, individual images differed from one another in the spatial configuration of their pain response but were matched on net response. Although global intensity information may be task related, it is also affected in idiosyncratic ways by preprocessing decisions, image acquisition parameters, and spurious physiological noise. We also z-scored pain ratings within study to partially control for quantitative differences in the rating scales and calibrations used (S1 Fig). This preserved between-participant differences within-study.

Standardized imaging data and pain ratings were averaged within the pain rating quartiles for each participant. This removed some portion of the overall variance in our pain reports, and all reported $r^2$ statistics should be interpreted as variance explained in quartiled data, but this was necessary for computational tractability. Quartiled data in particular was selected rather than some other quantile interval (e.g., tertiles or quintiles) because it has been established as a convention by previous studies [34] since it is the minimum level needed for all measures to contribute to estimates of linear intensity effects. With tertiles, e.g., the middle quantile only affects an intercept estimate, while the intensity effect is fully determined by the lowest and highest quantiles.

Incidentally, quartiling also denoised our data, partially aligning variability in our brain data with our outcome variable (pain rating), improved PCR models by increasing the sensitivity of principal component analysis to pain relevant features and reduced noise related dilution of regression parameter estimates (see MVPA: General method description below). Thus, we expect this approach to have more accurately estimated underlying generative processes

relative to single trial derived models. This nominal impact on our performance metrics has no impact on our conclusions because all our model comparisons were performed as repeated measures analyses, and each model performance metric was affected equally.

## MVPA: General method description

Multivariate patterns were fit to each modeling space using PCR with dimensionality determined by Bayesian optimization of mean squared error (MSE, S2 Text). The dimensionality search space spanned 1-R PCA dimensions, where R is the matrix rank of the training data. We precluded intercept only models (0 PCA dimensions) a priori. Finally, we used 2 × 5-fold repeated k-fold nested CV to estimate model performance. The inner CV loop selected PCR dimensions (S2A Fig), the outer CV loop obtained properly cross-validated model predictions (S2B Fig), and repetition stabilized results with respect to idiosyncrasies of fold partitions. We evaluated cross-validated predictions using within-participant Pearson correlation between predicted and observed pain ratings. This means the scope of this study is limited to within-participant variance in pain rather than between individual differences. Participant-wise Pearson *r*-values were normalized by Fisher's *z* transformation (hereon: "within-participant correlations"). PCR implementation is available at https://github.com/bogpetre/mlpcr0.

## MVPA multistudy analysis and model generalization performance

We pooled data across 7 studies and trained separate MVPA models for each modeling space. Pooling individual participant data from multiple studies to fit a single model is commonly referred to as mega-analysis [93,94], which has been shown to perform better than 2-stage procedures where models are fit separately for individual studies before being integrated in a meta-analytic model [55]. All input spaces required estimation of a PCR dimension. We estimated these dimensions in a nested CV loop as described in the "General method description" above; however, modular spaces (elementary regions, cRSN, and fRSN) required the additional selection of a best module. We used an additional level of CV for this and sampled modules exhaustively (so counting the nested 5-fold CV for optimization of PCR dimension within each area this was ultimately a 3× nested CV).

Model comparisons were performed using mixed effect models of within-participant correlations, with random participant and random study intercepts and random study slopes. Orthogonal contrast codes tested planned comparisons between within-participant correlations across models, similar to an rm-ANOVA but with additional random study effects. Null hypothesis tests were conducted for all nonsignificant contrasts using BFs (Bayesian hypothesis tests in S2 Text).

The multistudy analysis balanced participants across studies. Our smallest study had 16 participants, resulting in random sampling of 16 participants from each study, for a total of 112 participants and 6,668 stimulus trials. After quartiling data, this resulted in model training over 448 observed points.

## Study-wise MVPA and learner generalization performance

We defined the learner as the combination of algorithm (PCR) and the modeling space (regional, cRSN, fRSN, pain pathways, Neurosynth, or full brain). For learner performance estimates, we did not aggregate data across studies, instead defining 7 nonoverlapping datasets, one for each study (using the complete participant sample available in each case, in contradistinction with the multistudy analysis where participants were balanced across studies), and trained separate patterns for each. This study-wise MVPA used a total of 171 participants and 9,773 trials (16 to 30 participants, 256–2592 trials per study MVPA, Table 2). As for the

multistudy analysis, we averaged data within-pain rating quartile within participant and evaluated the performance of MVPA patterns using $2 \times 5$ repeated k-fold nested CV. Our statistical models were the same as for the multistudy analysis: mixed effect models of within-participant correlation of predicted versus observed pain.

## Supporting information

**S1 Text. Supporting information results.**
(DOCX)

**S2 Text. Supporting information methods.**
(DOCX)

**S1 Fig. Histograms of single trial pain reports across studies.** Data shown are unstandardized, and, instead, the x-axis was scaled to match the rating scale range offered to participants. Underlying data: https://github.com/canlab/petre_scope_of_pain_representation/tree/main/figureS1.
(TIF)

**S2 Fig. We tested 6 different hypotheses regarding the scope of pain representation using cross-validated PCR to predict pain intensity. (A)** We used PCR to learn a model from an area. We estimated model performance using 5-fold CV, training models on data averaged within pain intensity quartiles (smallest squares) for each participant (4× squares and one color per participant). Participant data were not fragmented across folds (shown), and studies were balanced across folds (not shown). We estimated model performance using within participant Pearson correlation of predicted and observed ratings. **(B)** The model fitting algorithm selected optimal model hyperparameters to minimize MSE, which it estimated using nested CV folds. In the case of multiarea hypotheses (A, top row), the algorithm treated region selection as a hyperparameter, and identified a single best region for each outer fold (folds illustrated in B). This region may have differed across outer CV folds. PCR dimensionality was optimized by estimating expected MSE in an additional innermost CV loop (3 levels of nested folds). In the case of distributed area hypotheses (A, bottom row), the algorithm only performed the latter step (2 levels of nested folds). PCR hyperparameter optimization folds are not illustrated. CV, cross-validation; MSE, mean squared error; PCR, principal component regression.
(TIF)

**S3 Fig. Distributed models are more pain specific than local models.** In validation studies where participants rated aversiveness of sound or warmth of nonpainful noxious heat as well as heat pain, models on average predicted nonpain ratings (left, warmth prediction $p = 3\text{e-}4$, sound prediction $p = 0.001$), and specifically fRSN, cRSN, and neurosynth models predicted warmth ($p = 1.2\text{e-}3$, $4.2\text{e-}3$, and $9.8\text{e-}4$ resp), while the pain pathways model predicted both warmth and sound ratings ($p = 1.7\text{e-}3$, $p = 2.7\text{e-}4$, resp). However, models predicted pain ratings better than not-pain ratings (center, pain > warmth, $p = 0.008$, pain > sound $p = 0.037$). Accuracy of pain rating predictions are indirectly indicated by net model response differences (right), which likewise showed greater responses for pain than not-pain stimuli in a manner roughly proportional to the intermodal within-participant differences in correlations (center). *Holm–Sidak $\alpha = 0.05$ for 12 comparisons. Underlying data: https://github.com/canlab/petre_scope_of_pain_representation/tree/main/figureS3. cRSN, coarse resting-state network; fRSN, fine resting-state network.
(TIF)

**S1 Table. Study behavioral factors overview.** Regression model coefficients (Fig 1B) shown. Standardized coefficients (SEM), regression of quartiled pain report with (unlisted) participant fixed effects. $p < 0.05$, all effects shown. ns, nonsignificant, excluded from model. blank cell, invariant or absent factor. [†,‡]Unique unlisted nonsignificant factors ([†]masked emotional facies, [‡]cognitive self-regulation).
(XLSX)

**S2 Table. Model and learner performance across modeling spaces (Fig 5A and 5B).** [†]cross-validated model performance (Pearson-r) within-participant is normalized by z-fisher transformation and modeled using a mixed effects design (random participant intercept, random study intercept and slope). Inverse z-Fisher transformed model coefficients and CI shown. CI0.95, 95% confidence Intervals. df, degrees of freedom estimated using Satterthwaite's correction.
(XLSX)

**S3 Table. Contrasts of model and learner performances across modeling spaces (Fig 5A and 5B).** [†]Performance contrasts are modeled using a mixed effects design (random participant intercept, random study intercept, and slope). [‡]z-fisher transformation of cross-validated model performance (within-participant Pearson-r). df, degrees of freedom estimated using Satterthwaite's correction. NS, neurosynth; PP, pain pathways. [*]$p < 0.05$.
(XLSX)

**S4 Table. Mediation model (standardized) coefficients and 95% confidence intervals (Fig 6).** [*,†]$p < 0.05$, bias-corrected bootstrap confidence interval, 5,000 repetitions. Factors included based on statistical significance in multiple regression on pain ($t$ test, $p < 0.05$). All parameters and statistics estimated while modeling participant fixed effects (not shown).
(XLSX)

**S5 Table. Mediation effects for multistudy models (Fig 5A), product of standardized coefficients, [95% confidence] and (mediation partial $R^2$).** [*]$p < 0.05$, bias-corrected bootstrap confidence interval, 5,000 repetitions. Factors included based on statistical significance in multiple regression on pain ($t$ test, $p < 0.05$). All parameters and statistics estimated while modeling participant fixed effects (not shown).
(XLSX)

**S6 Table. Mediation analysis of pooled study specific models' predictions, product of standardized coefficients, [95% confidence] and (mediation partial $R^2$).** [*]$p < 0.05$, bias-corrected bootstrap confidence interval, 5,000 repetitions. Factors included based on statistical significance in multiple regression on pain ($t$ test, $p < 0.05$). All parameters and statistics estimated while modeling participant fixed effects (not shown).
(XLSX)

## Acknowledgments

Data storage was supported by the University of Colorado Boulder "PetaLibrary." We would like to thank Dr. Christian Büchel for contributing data to this project and Dr. Marta Čeko for comments and feedback on the manuscript.

## Author Contributions

**Conceptualization:** Bogdan Petre, Philip Kragel, Tor D. Wager.

**Data curation:** Bogdan Petre, Lauren Y. Atlas, Stephan Geuter, Marieke Jepma, Leonie Koban, Anjali Krishnan, Marina Lopez-Sola, Elizabeth A. Reynolds Losin, Mathieu Roy, Choong-Wan Woo.

**Formal analysis:** Bogdan Petre, Lauren Y. Atlas, Stephan Geuter, Marieke Jepma, Leonie Koban, Anjali Krishnan, Marina Lopez-Sola, Elizabeth A. Reynolds Losin, Mathieu Roy, Choong-Wan Woo.

**Funding acquisition:** Tor D. Wager.

**Investigation:** Bogdan Petre.

**Methodology:** Bogdan Petre, Tor D. Wager.

**Project administration:** Bogdan Petre, Tor D. Wager.

**Resources:** Stephan Geuter, Tor D. Wager.

**Software:** Tor D. Wager.

**Supervision:** Tor D. Wager.

**Validation:** Bogdan Petre.

**Visualization:** Bogdan Petre.

**Writing – original draft:** Bogdan Petre, Philip Kragel, Tor D. Wager.

**Writing – review & editing:** Bogdan Petre, Philip Kragel, Tor D. Wager.

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
