## [Editor Report · Decision Letter 0]

8 Jul 2021

Dear Dr Wager, 

Thank you for submitting your manuscript entitled "Evoked pain intensity representation is distributed across brain systems: a multistudy megaanlysis" for consideration as a Research Article by PLOS Biology.

Your manuscript has now been evaluated by the PLOS Biology editorial staff and I am writing to let you know that we would like to send your submission out for external peer review.

Please re-submit your manuscript within two working days, i.e. by Jul 12 2021 11:59PM.

Kind regards,

Lucas Smith

Associate Editor

PLOS Biology

lsmith@plos.org

---

## [Decision Letter · Decision Letter 1]

28 Sep 2021

Dear Tor,

Thank you for submitting your manuscript "Evoked pain intensity representation is distributed across brain systems: a multistudy megaanlysis" for consideration as a Research Article at PLOS Biology, and thank you again for your patience during the protracted review process. Your manuscript has now been evaluated by the PLOS Biology editors, an Academic Editor with relevant expertise, and by several independent reviewers. You can find the reviewer comments, as well as a comment from the Academic Editor below my signature.

As you will see, while each reviewer has called aspects of the study interesting, they have raised a number of important concerns that preclude publication of the study in its current form. Reviewer 2 has commented that the manuscript is very long and at times, difficult to understand. Reviewer 1 and 3 have each noted inaccuracies with the framing of the study, which will need to be carefully addressed to ensure that the data and state of the field are accurately represented in the manuscript. Similarly, the Academic Editor has identified a claim made in the abstract that s/he feels is not fully supported. The reviewers have also identified a number of important points that require further clarifications and analyses. 

Additionally, we note that Reviewer 2 has commented that the paper lacks a clear conceptual advance and questions its suitability for PLOS Biology. After careful discussion within the team and with the Academic Editor, we think that the study is, in principle, within the scope of a PLOS Biology article, and we do not share these specific concerns about the conceptual advance. 

Therefore, while we will not be able to accept the current version of the manuscript, we would welcome re-submission of a much-revised version that takes into account the reviewers' comments. We think that to be considered further, in addition to addressing the technical concerns raised by the reviewers, the revised manuscript would need to be substantially re-written to make it more streamlined and accessible to a wide readership. IMPORTANT: Please note, that for systematic reviews and meta-analyses, we require that submissions adhere to the PRISMA statement and include a completed PRISMA checklist and flow diagram. Blank templates of the checklist and flow diagram can be downloaded from the PRISMA web site. As you revise your study, please make sure to meet these requirements. You can find more information on our policy here: https://journals.plos.org/plosbiology/s/best-practices-in-research-reporting

We cannot make any decision about publication until we have seen the revised manuscript and your response to the reviewers' comments. Your revised manuscript is also likely to be sent for further evaluation by the reviewers. Given that the reviewers think that parts of the manuscript are currently difficult to understand, such a revision may lead to new concerns being identified, which we would need to take into account.

We expect to receive your revised manuscript within 3 months. 

**IMPORTANT - SUBMITTING YOUR REVISION**

*Re-submission Checklist*

*Published Peer Review*

*PLOS Data Policy*

*Blot and Gel Data Policy*

Sincerely,

Lucas Smith

Associate Editor

PLOS Biology

lsmith@plos.org

REVIEWS:

COMMENTS FROM THE ACADEMIC EDITOR:

The authors make the assertion in the abstract 'These findings quantify the extent that representation of evoked pain experience is distributed across multiple cortical and subcortical systems'. But this is actually false, because it conflates the ability to predict experience with experience itself (i.e. it doesn't show that this representation is necessary or sufficient for experience). It should read 'the representation of evoked responses that can be used to predict pain experience', which is fundamentally different.

Reviewer #1, Jian Kong: In this manuscript, the authors compared multivariate predictive models to investigate the spatial scale and location of acute pain intensity representation. This is a very interesting study; the analysis is appropriate, and the manuscript is well-written and organized. Please see my following comments, which hopefully can improve the manuscript.

1. As the authors mentioned, distinct representations have been demonstrated for different types of pain including somatosensory and vicarious pain, and different kinds of clinical pain. It seems to the reviewer that this study has been focused on the experimental heat pain. I would suggest the authors change the title, abstract and other related sections to be clarify / emphasize this point. 

2. Please specify the pain intensity rating ranges for each study. Can these models predict mild-moderate pain ratings better than intensive pain ratings or vice versa?

3. If the reviewer is correct, a 0 - 10 VAS scale was used to evaluate the pain intensity in these studies. One question is if transforming continuous pain intensity rating into a category scale (for instance dividing the pain intensity into mild, moderate, and strong), will that change the results significantly? A related question is do the authors believe that the brain codes the pain intensity as a continuous variable or a category variable in the real world? 

Reviewer #2: Thè manuscript by Petre and colleagues describes the use of machine learning for the decoding of pain intensity across different studies. This is an interesting but very long and complex manuscript. I see no major issue with this study but do not see any fundamental advance compared to the previous machine learning application on the neurological pain signature from the authors. Such an incremental advance is usually regarded as insufficient for journals of the likes of Plos Biology.

Some recommendations:

The manuscript is very long, difficult to understand, and asks a lot from the average scientific reader in pain research. The introduction contains some redundant information, e.g. the first paragraph.

To shorten the manuscript I would suggest the authors focus on the essential aspects of the study. My suggestions

move analysis steps before ML to supplement as they are standard procedures. 

Shrink introduction

A more focussed discussion. 

The results part is difficult to read. I understand that the manuscript structure of Plos Biol with a closing methods section requires a more extended results part. However, understanding the results without having read the methods section is not doable for this complex study anyway.

The methods section would benefit from a bit more structure.

I do not fully understand why it is not possible to use a leave-one-out approach for all 11 studies. In my view this can be done and is less prone to subjective division between training data and application data.

I would strongly recommend the authors to drop MEGA. The number of included studies and participants is nowhere near MEGA. It just sounds [...]. The senior author may consider that this terminology would be mostly bound to the first author, which could be embarrassing for him in his future scientific career (I know what I'm talking about).

I wonder if any of the co-authors would understand the study in full detail. I do not feel competent enough to evaluate the methods due to its complexity. I have no idea how the analyses in the supplement relate to the analyses in the main document. I guess there are more pain researchers who would find this manuscript incomprehensive and unreadable. 

Reviewer #3: The article from Petre and colleagues presents a novel mega-analysis of fMRI studies using multivariate pattern analysis methods to consider the performance of different models - each with its own interesting anatomical relevance. The methods are novel and permit an interesting window into scale of pain processing in the brain using high quality data with enhanced sensitivity offered by the mega-analysis. The analysis of specificity with non-noxious data, and the external validation results, lead me to be encouraged and convinced by the approach. However, I have a few comments and queries for the authors about aspects of the manuscript, specifically some decisions on interpretation and methods employed. 

1. In the abstract, and (to a lesser degree introduction) the authors imply that the scale of representation of pain is controversial. I am not sure about this. Possibly it is own bias, but I believe most researchers are convinced by the fact that pain is present across cortical-subcortical systems. Perhaps the controversy referred to comes from recent papers that argue for a salience explanation, but it reads a little as though there is a genuine debate with a body of scientists arguing for localised modular centres of pain processing. Whilst this may have been true historically, I don't think it is the case now and would prefer to present this work as an interesting and valid comparison of models, rather than the resolution of an ongoing debate. 

2. The decision making for organising the data into quartiles is not very clear to me. This is important because it can clearly affect the results, but it comes to us as a fairly arbitrary decision. I seethe computational explanation, but why not tertiles or 5ths which would be similarly feasible? It would be good for the reader to understand the decision making here. 

3. The value of some mediation analyses is not immediately clear to me. The first mediation we are presented with seems like a reasonable attempt to understand how the brain models capture experimental manipulation in the data. I see that this can enhance interpretation, but why utilise only the full brain model? I can see some argument for using the other distributed models which according to some measures outperformed full-brain, or perhaps using mediation to tease apart differences in performance of different models. To inly use the full model is strange to me. 

4. To follow on from the previous point, I found the mediations in the learner generalized performance section to be problematic. Firstly, it required some pooling of data across studies which are quite different, a problem that the authors mention (because it limits their interpretation) at the end of this section themselves. In a paper that is quite dense, I am not convinced of the benefit or need for these analysis.

5. I realise this is quite a long paper, but I really feel that a very brief explanation of the anatomical underpinnings of the models (particularly those with anatomical relevance such as PP and RSN regions) should be mentioned in the text of the article, rather than just supplemental. The detail can still be signposted in supplemental. 

6. On Figure 4, I found the text around wheel plot figures to be quite difficult, and the grey lines that came off some labels to lead to the brain maps were also a little problematic. It is a minor thing but I think these bits could be a little clearer. 

Reviewer #4: This is an excellent mega-analysis investigating whether pain representation is best characterized as localized, multi brain system, or whole brain. The authors provide a detailed and transparent description of their approach and results from this complex analysis. The results will have an impact on future planning of studies examining the representation of pain in the brain. Great contribution and wonderful and explanatory figures.

---

## [Decision Letter · Decision Letter 2]

14 Mar 2022

Dear Tor,

Thank you for submitting your revised Research Article entitled "Evoked pain intensity representation is distributed across brain systems: A Multistudy Analysis" for publication in PLOS Biology. I'm handling your paper for the final stages, as my colleague Luke Smith is currently on parental leave. I've now obtained advice from the original reviewers and have discussed their comments with the Academic Editor. 

Based on the reviews, we will probably accept this manuscript for publication, provided you satisfactorily address the following data and other policy-related requests.

a) Please flip the title around to give us an active verb and avoid punctuation: "A multistudy analysis reveals that evoked pain intensity representation is distributed across brain systems"

b) Please provide a blurb according to the instructions in the submission form.

c) As this is a meta-analysis, we draw your attention to our relevant guidelines here: https://journals.plos.org/plosbiology/s/best-practices-in-research-reporting#loc-reporting-guidelines-for-specific-study-types Reports of systematic reviews and meta-analyses must adhere to the PRISMA statement as a guide, and include a completed PRISMA checklist and flow diagram to accompany the main text. Authors must also state within their Methods section whether a protocol exists for their systematic review, and if so, provide a copy of the protocol as Supporting Information.

d) Please address my Data Policy requests below; specifically, we need you to supply the numerical values underlying Figs 1B, 2, 3, 4, 5AB, 6, 7AB, 8, S1, S2AB, S3. Please also cite the location of the data clearly in each relevant main and supplementary Fig legend, e.g. “Data underlying this Figure can be found in S1 Data”.

We expect to receive your revised manuscript within two weeks. 

*Published Peer Review History*

*Press*

Best wishes,

Roli

Senior Editor,

rroberts@plos.org,

PLOS Biology

DATA POLICY:

We note that your "raw" data are available in Github and Neurovault. However, we also ask that all numerical values summarized in the figures and results of your paper be made available in one of the following forms:

Regardless of the method selected, please ensure that you provide the individual numerical values that underlie the summary data displayed in the following figure panels as they are essential for readers to assess your analysis and to reproduce it: Figs 1B, 2, 3, 4, 5AB, 6, 7AB, 8, S1, S2AB, S3. NOTE: the numerical data provided should include all replicates AND the way in which the plotted mean and errors were derived (it should not present only the mean/average values).

DATA NOT SHOWN?

REVIEWERS' COMMENTS:

Reviewer #1:

[identifies himself as Jian Kong]

Congratulations! Jian Kong

Reviewer #2:

The authors have sufficiently answered my comments.

Reviewer #3:

I believe that this paper is greatly improved by the changes that authors have made. I am happy to see the new language in reference to 'ongoing debate' which I think frames this work in a more coherent fashion. The mediation analysis, which I previously suggested for removal, is hugely improved in the enhanced explanation in the ms. I now believe that this section offers the reader important additional insight and warrants it's place, this is (IMO) a huge improvement in the manuscript as a whole. 

I believe that this work is important, novel and relevant and I would strongly suggest that it is suitable for publication.

---

## [Editor Report · Decision Letter 3]

7 Apr 2022

Dear Tor,

On behalf of my colleagues and the Academic Editor, Ben Seymour, I'm pleased to say that we can in principle accept your Research Article "A multistudy analysis reveals that evoked pain intensity representation is distributed across brain systems" for publication in PLOS Biology, provided you address any remaining formatting and reporting issues. These will be detailed in an email that will follow this letter and that you will usually receive within 2-3 business days, during which time no action is required from you. Please note that we will not be able to formally accept your manuscript and schedule it for publication until you have completed any requested changes.

Sincerely, 

Roli

Roland G Roberts, PhD 

Senior Editor 

PLOS Biology

rroberts@plos.org